# Potential of Newly Synthesized Sea Buckthorn Phytocarriers as Anti-Inflammatory Active Agents

**DOI:** 10.3390/ph18020212

**Published:** 2025-02-05

**Authors:** Ionela Daniela Popescu, Elena Codrici, Sevinci Pop, Tudor Emanuel Fertig, Maria Dudău, Iliuta Laurentiu Anghelache, Nicoleta Constantin, Radu Marian Marinescu, Vlad Mihai Voiculescu, Georgiana Ileana Badea, Mirela Diaconu, Monica Elisabeta Maxim, Mihaela Scurtu, Kliment Zanov, Ana-Maria Enciu, Simona Carmen Litescu, Cristiana Tanase

**Affiliations:** 1Victor Babes National Institute of Pathology, 99-101 Splaiul Independentei, Sector 5, 050096 Bucharest, Romaniaspop@ivb.ro (S.P.); emanuel.fertig@umfcd.ro (T.E.F.); laurentiu.anghelache@ivb.ro (I.L.A.); cristianatp@yahoo.com (C.T.); 2Faculty of Medicine, Carol Davila University of Medicine and Pharmacy, 8 Eroilor Sanitari, 050047 Bucharest, Romania; radu.gh.marinescu@stud.umfcd.ro (R.M.M.);; 3Centre of Bioanalysis, National Institute of Research and Development for Biological Sciences, 296 Independenței Bd., District 6, 060031 Bucharest, Romania; 4“Ilie Murgulescu” Institute of Physical Chemistry, Romanian Academy, 202 Splaiul Independentei, 060021 Bucharest, Romania; mmaxim@icf.ro; 5Cromatec Plus SRL Str. Petre Ispirescu nr. 1, Sat Tancabesti, Comuna Snagov, 077167 Ilfov, Romania; 6Cajal Institute, Titu Maiorescu University, 22 Dâmbovnicului, Sector 4, 040441 Bucharest, Romania

**Keywords:** sea buckthorn ethanolic extract, nanocarriers, phytocarriers, anti-inflammatory effect, IL-8

## Abstract

**Background:** Phytocarriers are advanced drug delivery systems that use biocompatible and biodegradable materials to enhance the efficacy, stability, and bioavailability of natural products. The sea buckthorn (*Hippophae rhamnoides *L.) berry extract is rich in essential fatty acids and antioxidants, including vitamin C, vitamin E, and anthocyanins, which contribute to its wide-ranging health benefits. In this study, we assessed the morphology, intracellular delivery, and anti-inflammatory effect of sodium cholate (NaC) and sodium deoxycholate (NaDC)-based phytocarriers loaded with ethanolic extract from sea buckthorn berries (sea buckthorn carrier nanostructures, further defined as phytocarriers). **Methods:** Negative and electron cryo-microscopy were used to analyze hollow and loaded nanocarriers. The cyto-compatibility of nanocarriers was assessed by endpoint (LDH and MTS) and real-time cell assays, on both human fibroblasts (HS27) and human normal monocytes (SC). The anti-inflammatory effect of hollow and loaded nanocarriers was tested by multiplexing. **Results:** The negative and electron cryo-microscopy analyses showed that NaC-based phytocarriers were spherical, whilst NaDC-based phytocarriers were predominantly polymorphic. Moreover, the NaDC-based phytocarriers frequently formed large lipid networks or “plaques”. Although 24 h cytotoxicity testing showed both types of nanocarriers are biocompatible with human fibroblasts and monocytes, based on a long-term real-time assay, NaDC delayed fibroblast proliferation. NaC sea buckthorn phytocarriers did not impair fibroblast proliferation in the long term and they were uptaken by cells, as shown by hyperspectral microscopy. NaC nanocarriers and NaC sea buckthorn phytocarriers induced an anti-inflammatory effect, lowering IL-8 cytokine production in normal human monocytes as soon as 4 h of treatment lapsed. **Conclusions:** NaC-derived phytocarriers loaded with sea buckthorn alcoholic extract are a cell-compatible delivery system with anti-inflammatory properties.

## 1. Introduction

Phytocarriers are advanced drug delivery systems that use biocompatible and biodegradable materials to enhance the efficacy, stability, and bioavailability of natural products [1]. Depending on the structure and formulation method, nanocarriers are classified into several categories, whose number increases with technological development. Most are organic, such as the widely used lipid-based nanocarriers (liposomes, microemulsion drug delivery systems, nanostructured lipid carriers, solid lipid nanoparticles, transferomes, and ethosomes), but also polymeric nanocarriers (in the nanoparticles, micelles, and dendrimers). There are also inorganic and hybrid nanocarriers. Some novel nanocarriers are niosomes, liquid crystals, and nanotubes [2].

One natural product benefiting from nanocarrier delivery is the sea buckthorn extract, derived from the berries, seeds, and leaves of the *Hippophae rhamnoides* plant. The berry extract contains a significant variety of essential fatty acids and antioxidants, including vitamin C, vitamin E, and polyphenols, which contribute to its wide-ranging health benefits and applications [3].

Phytocarriers (the structure formed between the natural compound extract and its nanocarrier) offer several benefits over conventional drug delivery methods as follows: 1. enhanced bioavailability; 2. controlled release, providing a steady therapeutic effect over a longer period [4]; 3. targeted delivery, as phytocarriers can be engineered to deliver the sea buckthorn extract specifically to the desired site of action, thereby reducing side effects and increasing efficacy; and 4. improved stability, protecting the extract from environmental factors such as light, heat, and oxygen [5]. Studies have demonstrated that these nanocarriers enhance the bioavailability of the extract’s active components, making it more effective in treating conditions such as arthritis, skin disorders, and even certain types of cancer [6,7]. Additionally, the antioxidant and antimicrobial properties of the encapsulated extract enhance its value as a food supplement, nutraceutical, and cosmeceutical, further broadening its applicability [8].

Several types of nanocarriers have been developed for the delivery of sea buckthorn extracts, each with specific advantages. Liposomes are spherical vesicles consisting of lipid bilayers capable of entrapping both lipophilic and hydrophilic compounds. They are recognized for their safety profile and have been used to enhance the delivery of sea buckthorn extract by protecting its active components and ensuring controlled release [7]. Nanostructured lipid carriers (NLCs) are an advanced type of lipid-based carrier with improved stability and controlled release properties. The encapsulation of sea buckthorn extract into NLCs has been shown to enhance its antioxidant and antimicrobial properties, making it a promising candidate for therapeutic applications [8]. Polymeric nanoparticles, synthesized from biodegradable polymers, have been used to encapsulate various natural products, including sea buckthorn extract. These carriers offer the benefits of high drug loading capacity, controlled release, and targeted delivery, which are crucial for improving the therapeutic outcomes of encapsulated bioactive compounds [5]. Solid lipid nanoparticles (SLNs) are another lipid-based carrier system known for their stability and ability to encapsulate both lipophilic and hydrophilic substances. They have been employed to enhance the bioavailability of sea buckthorn extract and to protect its sensitive compounds from degradation [9]. The long-term safety and regulatory approval of these nanocarrier systems are still under examination, requiring more comprehensive research and clinical trials [5,9].

Despite these advancements, several issues remain to be solved. Unlike small molecule drugs, phytocarriers presents different challenges related to cytotoxicity testing due to their large size and polarity. Distinct from conventional drugs, some nanocarriers are unable to diffuse through the cell membrane and instead enter cells via endocytosis [9]. There is still a limited understanding of the mechanisms of action and potential toxicity of these drug delivery systems, which requires supplementary studies. Additionally, long-term safety profiles of these nanocarriers need comprehensive evaluation to ensure their viability for clinical use [10]. Recent research efforts are increasingly focusing on the synthesis of nanocarriers through environmentally safe chemical reactions, incorporating plant extracts [11]. As this field advances, it will be crucial to address these challenges to improve the translational value of current nanomedicine research and to fully realize the therapeutic potential of sea buckthorn extract-loaded nanocarriers.

The aim of this study is to produce evidence that nanocarriers, loaded with sea buckthorn extract originating from Romania, are suitable for further potential development as food supplements with health/therapeutic benefits.

## 2. Results

In order to assess the utility of sodium cholate (NaC) and sodium deoxycholate (NaDC) edge activator nanocarriers for the nanodelivery method, morphological analysis, cytotoxicity and proliferation assays, as well as cellular uptake were investigated. Both nanocarriers, hollow spherical structures loadable with the cargo of choice (further referred to as “hollow nanocarriers”) and nanocarriers loaded with sea buckthorn extract (further referred to as “phytocarriers”), were used in the analysis, and the anti-inflammatory effect of the sea buckthorn ethanolic extract was determined by multiplexing.

### 2.1. NaC-Derived Phytocarriers Are Spherical, Loadable, and More Homogeneous than NaDC Phytocarriers

The morphology of both hollow and loaded nanostructures was assessed using NS-TEM and cryo-TEM.

NaC-based nanostructures were relatively homogenous, with diameters ranging from 20 to 35 nm and averages of 28.6 nm in NS-TEM specimens (n = 50, Figure 1a) and 25.8 nm in cryo-EM (n = 40, Figure 1b). Particles larger than 50 nm were absent from the samples analyzed. While, in NS-TEM, particles frequently presented a “cup-shape” morphology with a collapsed center (Figure 1a), in cryo-EM, they appeared spherical due to their frozen-hydrated state (Figure 1b). By contrast, NaDC nanocarriers were frequently polymorphic in NS-TEM and only rare spherical particles were observed (Figure 1c). As these nanostructures appeared as isolated spheres in cryo-EM (Figure 1d), the polymorphism seen in NS-TEM was likely due to increased aggregation and/or fusion during sample dehydration. As compared to NaC nanostructures, NaDC particles were larger, ranging from 29 to 54 nm (average 44.5 nm, n = 40). In cryo-TEM, NaDC nanocarriers appeared more electron-dense (darker) than NaC nanocarriers, suggesting they have a different structural lipid organization.

Similarly to NaC hollow nanocarriers, in cryo-TEM, NaC phytocarriers appeared relatively homogenous in terms of size, with an average diameter of 26.9 nm (min 19, max 41, n = 40) and most particles ranging between 22 and 25 nm. The second most abundant class of particles had diameters below 20 nm. In NS-TEM, occasional larger phytocarriers could be observed (40 to max 240 nm). The absence of these larger particles in cryo-EM preparations can be explained by the differences in particle surface charges, which would favor the deposition of larger phytocarriers to the carbon surface of the grid rather than the holes with vitreous ice. Regardless of size, NaC phytocarriers were spherical in both NS-TEM (Figure 1e) and cryo-TEM (Figure 1f). By contrast, NaDC phytocarriers were predominantly polymorphic in NS-TEM, frequently forming large lipid networks or ”plaques” (Figure 1g). In cryo-TEM, NaDC phytocarriers appeared as individual spheres, but with a very broad size distribution, from 17 to 85 nm (average 33.2 nm, n = 40). Some vesicles appeared electron-dense and larger vesicles presented smaller vesicles within, probably as a result of fusion events (Figure 1h).

The TEM morphological analysis therefore suggests that NaC phytocarriers and hollow nanocarriers are preferable to their NaDC equivalents due to their smaller size and homogeneity, which would facilitate cellular phagocytosis.

The obtained phytocarriers, either hollow or loaded, were assessed by Dynamic Light Scattering, zeta potential, and polydispersity index, as presented in Table 1.

Next, we analyzed the biocompatibility of the NaC and NaDC hollow nanocarriers and phytocarriers on the following two cell lines: human fibroblasts and human normal monocytes.

### 2.2. Biocompatibility Testing of Hollow Nanocarriers and Loaded Nanocarriers/Phytocarriers

Before cytotoxicity testing, the chemical composition of the extract was determined (Table 2). It should be mentioned that, to ensure easier traceability, the extract was normalized to epicatechin and rutin when the encapsulation efficiency was assessed.

The encapsulation efficiency was further assessed, as a percentage of the encapsulated compound reported to the total amount and was determined to be 75.62% for rutin and 82.44% for epicatechin.

The biocompatibility of the nanocarriers was assessed by endpoint cytotoxicity (LDH release) and cell viability assays (MTS), as well as by a real-time cellular proliferation assay.

The effect of two dilutions (1/50, 1/100) of NaC or NaDC loaded and hollow nanocarriers on cell viability and the possible induced cytotoxicity were tested by endpoint assays. Two types of cells were tested as follows: normal human fibroblasts, a generally accepted model for toxicity assays; and human normal monocytes, for the assessment of non-toxic concentrations for the subsequent inflammation-related assays (Figure 2a). After 24 h of treatment, the LDH release for fibroblasts was below 25% at all tested dilutions, which correlated with the percentage of viable cells (over 80%) for all analyzed samples. Similar results were obtained for the treated monocytes, where LDH release was below 20% at all tested dilutions, correlating with the cell viability results which were close to 100% for phytocarriers (loaded with extract) and over 80% for hollow nanostructures. None of the tested samples induced a significant increase in cell proliferation, as determined by endpoint assays for 24 h of treatment.

The endpoint cell proliferation assay was complemented by a real-time assay for cells adhesion and proliferation, where incubation time was increased, and data collection at intermediate times allowed for the calculation of the doubling times of treated cells.

Regarding NaC and NaDC hollow nanocarriers, no significant difference was observed between treated and non-treated fibroblasts in the first 48 h, consistent with the MTS analysis (Figure 2b). Extending the testing interval to 96 h revealed alterations in the cellular response profiles for both NaC and NaDC hollow nanocarriers at all concentrations, except for the NaC nanocarriers at a 1/100 dilution. Notably, a decrease in cellular index values was observed for NaDC nanocarriers at both dilutions and NaC nanocarriers at a 1/50 dilution. A 1/100 dilution of NaC nanocarrier-treated cells showed a similar proliferation curve as the untreated control for the whole period of time (96 h). A subsequent analysis of doubling time confirmed that NaC nanostructures at a 1/100 dilution do not modify the proliferation rate of normal human fibroblasts. Instead, for NaDC hollow nanocarriers at 1/50 and 1/100 dilutions and NaC at a 1/50 dilution, a stable and statistically significant increase in the doubling times of normal human fibroblasts, visible in real-time after approximately 50 h of incubation, can be observed (Figure 2b, right panel).

Regarding NaC and NaDC phytocarriers, we tested the response of fibroblasts in real time for up to 96 h. To analyze whether phytocarriers can modulate the cellular proliferation process and to avoid any confounding effects of different phytocarriers on initial cell attachment, the fibroblasts were allowed to adhere for 24 h before exposing them to the treatments. The results obtained for cells incubated with phytocarriers at a 1/50 dilution confirmed that human fibroblasts proliferate at lower rates than untreated control cells, an effect that is visible after approximately 24 h of treatment. The same is observed for the 1/100 dilution (Figure 2c, left panel). The decrease in proliferation rate can be observed after at least 24 h of incubation, and this is maintained throughout the entire experimental time. The doubling time analysis showed that, although the effect induced by the NaC phytocarriers is smaller than that induced by NaDC, it is still statistically significant when compared to the control (Figure 2c, right panel).

To conclude, NaC edge activator hollow nanocarriers do not modify fibroblast proliferation in either short or long term, confirming their effectiveness as biocompatible delivery vehicles, as opposed to NaDC-based nanocarriers. Interestingly, the sea buckthorn extract phytocarriers based on either NaC edge activator or NaDC edge activator, diluted 100-fold, induced an anti-proliferative effect on fibroblast cells when treated for longer periods of time.

### 2.3. NaC Nanocarriers and Sea Buckthorn-Loaded Phytocarriers Are Uptaken by Cells, Making Them Useful Vectors for Active Compounds Delivery

To answer the question of whether these carriers (hollow or loaded) were internalized by cells, hyperspectral imaging (HSI) libraries were acquired. These libraries were used to characterize the spectral signatures of the carriers and the cell background. By comparing these spectra, the intracellular presence of the carriers could be determined.

Both NaC and NaDC carriers were characterized using hyperspectral microscopy, but the cell-based analysis focused on the NaC nanocarriers. This decision was based on previous real-time data, as well as electron microscopy images, which demonstrated that NaC nanocarriers are superior in terms of morphologic homogeneity as well as biocompatibility to NaDC nanocarriers.

Hyperspectral data of hollow and loaded nanocarriers (phytocarriers) were measured in terms of the spectral mean (Figure 3a). No significant difference was noticed between hollow nanocarriers, while loading them with sea buckthorn extract induced a spectral shift to the right of the NaC-based phytocarriers (a maximum between 650 and 675 nm, compared to 610–630 nm). This spectral shift was used to identify the intracellular presence of a specific nanocarrier.

Fluorescence images were used to assess the uptake of NaC hollow nanocarriers and phytocarriers by macrophages. The phagocytic ability of macrophages was demonstrated using *E. coli* particles, which become fluorescent in an acidic environment (e.g., lysosomes) (Figure 3b). The NaC hollow nanocarriers and phytocarriers were uptaken by cells and were detectable after 24 h incubation (Figure 3c). The presence of hollow carriers and phytocarriers in treated macrophages was demonstrated by overlaying the hyperspectral data of each library over the microscopy images of phagocytosing cells.

To conclude, using hyperspectral microscopy, we demonstrated that both hollow and loaded NaC nanocarriers are uptaken by cells, hence their utility as intracellular delivery vehicles.

### 2.4. Pro-Inflammatory Cytokines Decreased in a Dose-Dependent Pattern After Nanocarrier Treatment

Next, the biological response of cells to selected hollow and loaded nanocarriers was tested from the following two points of view: whether the nanocarriers, loaded or not with sea buckthorn extract, induced inflammation themselves and, secondly, whether they can protect against inflammation. For the latter, cells were challenged with LPS and then treated with phytocarriers and hollow nanostructures. Dexamethasone treatment was used as a positive anti-inflammatory effect.

To assess whether the tested nanostructures induced a response from the exposed cells, normal human monocytes were incubated for 20 h and supernatant was harvested at 4 h (for early response) and 20 h (late response). Several pro- and anti-inflammatory cytokines were tested (IL-1beta, IL-2, IL-4, IL-6, IL-8, IL-10, IL-12p70, TNFa, TNFb, and INFg), but only IL-6 and IL-8 were detectable during 20 h of treatment. The early response was significant only for IL-8 expression, whereas IL-6 was detectable only at 20 h. Neither hollow nor loaded nanostructures induced, by themselves, an inflammatory response, highlighting their use as biocompatible carriers. In addition, when cells were challenged with LPS, both tested nanostructures induced a time-dependent anti-inflammatory response (Figure 4).

Treating cells for 20 h with LPS and 1/100 of NaC phytosomes led to significant decreases in IL-6 levels by 66% (ratio 0.34, *p* < 0.0001) and IL-8 by 54% (ratio 0.46, *p* < 0.0001). Additionally, treating cells with LPS and 1/100 concentrations of NaDC phytocarriers significantly reduced the levels of IL-6 (by 88.4% and 73%, respectively; *p* < 0.0001) and IL-8 (by 70% and 52%, respectively; *p* < 0.0001) (Figure 4). Similar results were also found when examining cells treated with hollow nanostructures and LPS compared to the LPS control group. Both IL-6 and IL-8 levels were significantly different between these groups (*p* < 0.0001).

## 3. Discussions

Incorporation of sea buckthorn extracts in lipid nanoparticles was previously described to successfully preserve the biological activity of the natural compounds [12,13]. Soy lecithin lipid nanoparticles were shown to be well tolerated by fibroblasts and monocytes after short-term incubation (24 h) [14]. Their proven lack of toxicity in the short term might account for why, in many studies, the cell viability was usually compared between the extract and the liposome-encapsulated extract at 24 h [15] or 48 h of treatment [16], while the nanocarrier was no longer tested. Our study also showed that, at 24 h, the hollow soy lecithin nanocarriers, as well as the sea buckthorn loaded nanocarriers, are well tolerated by human fibroblasts and monocytes.

One particular finding of this study is that, when assessed in real time for longer periods of time (96 h), both hollow nanocarriers, as well as sea buckthorn loaded nanocarriers, without being cytotoxic, decreased fibroblast proliferation. The effect was persistent for NaDC nanocarriers, even with increased dilution. In contrast, NaC nanocarrier-treated fibroblasts at a 1/100 dilution behaved similarly to the control. However, when loaded with sea buckthorn extract, even at a 1/100 dilution, the loaded NaC nanocarriers also decreased the proliferation of the fibroblasts. The ethanolic extract of sea buckthorn was previously reported to impair cell proliferation in reports mostly addressing tumor cells [17,18]. Sadowska et al. [19] noted an impaired fibroblast migration in the presence of non-polar components of the extract, while the phenolic component did not induce significant changes.

It has been previously shown that NaC and NaDC act differently in terms of micelle formation [20], which potentially impacts cellular uptake. Our study confirmed, through electron microscopy, that the type of edge activator used to generate nanocarriers influences their morphology and size. NaC-based nanoformulation performed better in the generation of spherical, possibly hollow nanocarriers, as assessed by electron microscopy. NaDC nanocarriers yielded elongated and ramified structures, which can potentially explain the inhibitory effect on cell proliferation observed in long-term real-time analysis. DLS data, the obtained negative zeta potential values, support the assertion that the obtained nanostructures are less subjected to aggregation in biological samples.

In terms of anti-inflammatory activity, the aqueous berry extract of sea buckthorn was shown to decrease the secretion of IL-6 and IL-8 in immune cells [21] and various epithelial cell models (CaCo_2_ [21], HaCaT [22]). For TNFα, the response depends on the cell model used—PMA-induced macrophages [23] respond differently than PBMC [21]. In animal models, the anti-inflammatory properties of sea buckthorn were exploited, especially in dermal applications [24] and intestinal pathologies [25]. Increasing the bioavailability of the active components found in sea buckthorn extracts can potentially bring additional benefits in pathologies characterized by systemic low-grade inflammation. This study also demonstrated the dose-dependent anti-inflammatory activity of both tested nanostructures, hollow or loaded, but the effect on cell proliferation should be taken into consideration. Correlated with the lack of toxicity of the 1/100 diluted nanocarriers, this study provided evidence for the delivery and anti-inflammatory effect of NaC nanocarriers and phytocarriers. Notably, NaC nanocarriers, by themselves, showed an anti-inflammatory effect, possibly attributable to phosphatidylcholine, a natural phospholipid known to have an anti-inflammatory effect, by the inhibition of the NF-kB pathway [26], including following LPS stimulation [27].

## 4. Materials and Method

### 4.1. Phytocarriers Preparation and Encapsulation

#### 4.1.1. Phytocarriers Preparation

Solutions of 24 mg of lecithin in 3 mL of 96% ethanol (8 mg/mL) were used to prepare hollow carrier nanostructures, further defined as hollow nanocarriers. Solutions of 24 mg lecithin in 3 mL sea buckthorn hydro-alcoholic extract (obtained by accelerated solvent extraction, ASE, in 96% alcohol and purified) were used to prepare nanostructures loaded with extract (further defined as phytocarriers). For the formation of the composite nano-architecture of phytocarriers, the minimum reaction time was 12 h under continuous stirring. This optimized time was used both for phytocarriers formation and for hollow nanocarriers preparations. After the 12 h, the high-pressure micro-homogenization of the nude lecithin and lecithin-phytoactive compound solution was performed, using 3 cycles of passage through Microfluidics LM20 at 20 kPa. Following this step, the homogenized solution was injected in very small volumes (smaller than 3 µL) of aqueous surfactant solution with edge activators (0.5% sodium cholate—NaC—or sodium deoxycholate—NaDC—and 0.5% Tween 80). In this case, the homogenized solution was either an alcoholic solution of lecithin or a lecithin solution with sea buckthorn extract in 96% alcohol. The injection of small volumes was performed under continuous stirring, at a ratio of 1:10 (v:v, 3 mL phospholipid-phytoactive compound solution to 30 mL aqueous solution with surfactants and edge activators). Finally, the 3 passages through Microfluidics LM20 at 20 kPa were repeated, resulting in nanostructures of lecithin in the composite nano-architectures (phytocarriers).

#### 4.1.2. Composition of Initial Sea Buckthorn Extract and Encapsulation Efficacy

The profile composition of the initial sea buckthorn extract was obtained by high performance liquid chromatography with diode array detection using a method previously developed by our group [28].

### 4.2. Cell Cultures and Treatments

Normal human fibroblasts Hs27 (ATCC CRL-1634™) were grown in Dulbecco’s Modified Eagle Medium (DMEM) with 4.5 g/L glucose, supplemented with 10% fetal bovine serum. Fibroblast cultures were grown and propagated in T-75 flasks, at 37 °C and 5% CO_2_ atmosphere. The cell culture medium was changed every 3 days. The cells at 80% confluence were passed 1/3. The tested solutions were diluted in the culture medium before the treatment. Untreated cells (control sample) refer to the normal cell medium supplemented with the vehicle in a corresponding concentration to the samples.

SC normal human monocytes (ATCC 9855) were routinely maintained in culture using IMDM medium, supplemented with 1% HT supplement and 0.1% 2-mercaptoethanol, according to the manufacturer’s protocol. To induce the macrophage phenotype, 25 ng/mL phorbol 12-myristate 13-acetate (PMA) treatment for 48 h was used, according to a previously published protocol [29]. The inflammation model was obtained by treatment with LPS (50 ng/mL) and dexamethasone (40 ng/mL) was used as an anti-inflammatory positive control.

### 4.3. Macrophage Phagocytosis Assay

The phagocytosis assay was performed on PMA-induced macrophages using fluorescently labeled *E. coli* particles (Invitrogen P35360), adhered on 8-well glass slides as described above (Nunc Lab-Tek #154526PK, ThermoFisher Scientific, Waltham, MA, USA). To test for liposome and phytosome phagocytosis, hyperspectral images associated with each type of carrier were plotted against live cell imaging. For ease of interpretation, a digital mask was applied to highlight the intracellular presence of particles.

### 4.4. Cytotoxicity and Viability Assay

#### 4.4.1. Cell Viability

Cell viability was measured after the cells’ exposure to hollow nanocarrier and phytocarrier suspensions by MTS tetrazolium compound [3-(4,5-dimethylthiazol-2-yl)-5-(3-carboxymethoxyphenyl)-2-(4-sulfophenyl)-2H-tetrazolium, inner salt] spectrophotometric test. A total of 15,000 cells/well were seeded in 96-well plates and treated for 24 h with two dilutions (1/50 and 1/100)) of NaC and NaDC phytocarriers. Following this timeframe, 20 µL of CellTiter 96^®^ AQueous One Solution Reagent (CellTiter 96^®^ Aqueous One Solution Cell Proliferation Assay, G3580, Promega, Madison, WI, USA) were subsequently added to each well, each of which contained 100 µL of fresh culture medium. Incubation was carried out at 37 °C for a duration of 3 h, in a humidified atmosphere containing 5% CO_2_. Cell metabolic activity, assessed by MTS reduction and formazan formation, was quantified by measuring absorbance at 490 nm on a Microplate Multimode Detector Zenyth 3100 (Anthos Labtec Instruments GmbH, Salzburg, Austria). Cell viability was determined as follows: viability (%) = 100 × [(experimental OD490) − (background mean OD490)]/mean absorbance at 490 nm of untreated cells.

#### 4.4.2. Cytotoxicity Assay

Cell membrane integrity was assessed by quantifying LDH (lactate dehydrogenase) release into the culture medium using the CytoTox 96^®^ assay (G1780, G1782, Promega). A maximum LDH release control was generated using lysis solution, in accordance with the manufacturer’s instructions. Following a 45 min incubation with lysis solution, plates were centrifuged at 600 rpm for 5 min. Subsequently, 50 μL of culture supernatant from each well was transferred to a new 96-well flat-bottom, clear plate, followed by the addition of 50 μL of CytoTox 96^®^ Reagent. Plates were incubated in the dark for 30 min, at room temperature. Subsequently, the reaction was stopped by adding 50 µL of stop solution. The absorbance was then measured at 490 nm using a Zenyth 3100 Microplate Multimode Detector (Anthos Labtec Instruments GmbH, Salzburg, Austria). We calculated cytotoxicity using the following formula: cytotoxicity (%) = 100 × [experimental LDH release (OD490) − background mean LDH release (OD490))/mean of maximum LDH release (OD490)].

#### 4.4.3. Real-Time Proliferation Assay

Normal fibroblasts HS27 cell adherence and proliferation were assessed using RTCA-DP xCelligence platform (Agilent). The cell adherence assay in the presence of the investigated nanostructures, hollow or loaded, in solvent solution of NaC and NaDC were performed at two dilutions as follows: 1/50 and 1/100. A number of 10,000 Hs27 cells were resuspended in cell culture media supplemented with carrier solutions, seeded in triplicates for each experimental condition, and monitored for 96 h. The experiments were repeated three times. The phytocarriers’ effect on cell proliferation was tested on cells which were left to adhere on E-plates for 24 h prior to cell treatment. After 24 h, the culture medium in each well was replaced by culture medium containing the NaC and NaDC phytocarrier solutions at the appropriate dilutions. Doubling times were calculated using RTCA software (Version 2.0).

### 4.5. Inflammatory Model in Monocytes/Macrophages

To assess the anti-inflammatory effects of the obtained nanostructures, phytocarrier 1 × 10^5^ monocytes were seeded per well in 24-well plates, pre-treated with 50 ng/mL LPS for 1 h, and then incubated with two dilutions (1/50, 1/100) of NaC and NaDC phytocarriers for 4 and 20 h. As part of the evaluation, we included a positive pro-inflammatory control (50 ng/mL LPS; L4391; Sigma Aldrich, Saint Louis, MO, USA; Merck KGaA, Darmstadt, Germany), a negative inflammation control (40 ng/mL dexamethasone sodium phosphate; E.I.P.I. Co., Tenth of Ramadan City, Egypt), and a combination of 50 ng/mL LPS with 40 ng/mL dexamethasone as a positive control for the anti-inflammatory effect. The cell culture supernatants were collected and stored at −80 °C until analysis.

### 4.6. In Vitro Quantification of Anti-Inflammatory Activity Using xMAP Technology

The analysis of the cell culture supernatants was performed using a multiplex bead-based assay (HCYTOMAG Milliplex MAP 10-plex, Millipore, Billerica, MA, USA) as per the manufacturer’s instructions. The beads (IL-1beta, IL-2, IL-4, IL-6, IL-8, IL-10, IL-12p70, TNFa, TNFb, and INFg) provided within the kit were incubated with buffer, cytokine standards, or samples in a 96-well plate at 4 °C, with 800 rpm orbital shaking o/n. Subsequent incubations with detection antibodies and Streptavidin Phycoerythrin Conjugate (SAPE) were performed in the dark with orbital shaking (800 rpm) and at room temperature. The Luminex 200 platform (Luminex Corp., Austin, TX, USA) was used for multiplex data acquisition, followed by data analysis with xPONENT 4.2 software. A 5-parameter logistic fit was used to generate calibration curves for each analyte. For each sample, duplicate analyses were performed, and the mean concentration was used for statistical analysis.

### 4.7. Transmission Electron Microscopy

Negative stain transmission electron microscopy (NS-TEM) and electron cryo-microscopy (cryo-TEM) were performed for the morphology assessment of isolated particles from samples of interest, following previously published methods [30]. Briefly, copper grids specific to each application (AGS160-3H, Agar Scientific, Rotherham, UK and R2/2, Quantifoil MicroTools, Großlöbichau, Germany, respectively) were glow-discharged before 3.5 μL of each suspension were deposited on the carbon-coated surface. Then, the grids were stained with 2.5% uranyl acetate for NS-TEM, or frozen for cryo-EM by rapid plunging in liquid ethane using a Leica EM GP system (Leica Microsystems, Wetzlar, Germany). For both sample preparation methods, image acquisition was carried out on a 200 kV Talos F200C, equipped with a Ceta 4k × 4k CMOS camera (Thermo Fisher Scientific, Waltham, MA, USA).

### 4.8. Fluorescence and Dark-Field Hyperspectral Microscopy

The acquisition of fluorescence and hyperspectral images (HSIs) was performed using Cytoviva Hyperspectral Imaging System equipped with a dual mode fluorescence (DMF) module (CytoViva, Auburn, AL, USA). Hyperspectral images were obtained from 20 μL of the phytocarriers suspension and a HSI acquisition with a 60× oil immersion objective. The HSIs were stored and analyzed using ENVI v.4.8 software (Harris Geospatial Solutions, Boulder, CO, USA). Phytocarriers’ spectral signature was collected using automatic particle filtration with an intensity ranging from 400 to 2000 units in order to remove unwanted spectra and save the selected ones in a spectral library. The identification of phytocarriers in samples containing cells was performed by the classification of the HSI to the saved spectral library using the spectral angle mapper. The spectral comparison of the phytocarriers (hollow or loaded) was performed after the normalization of the intensity and the calculation of the mean spectral analysis. The fluorescence images were acquired using the DMF module and QCapture Pro 7 software (QImaging, Surrey, BC, Canada).

### 4.9. Dynamic Light Scattering

Dynamic light scattering measurements were performed using a Zetasizer Nano ZS instrument (Malvern Instruments, Worcestershire, UK) with a detection angle of 173°. The surface charge (measured as Zeta potential) and polydispersity index were determined using the light scattering technique, according to the Smoluchowski model (in aqueous medium). All measurements were made at a temperature of 25 °C. The Zetasizer Nano software v3 was used for data processing.

### 4.10. Statistical Analysis

Statistical analysis was performed using GraphPad v7 data analysis software. One-way ANOVA test was used, with Dunnett’s multiple comparison post hoc analysis (* *p* < 0.05, ** *p* < 0.01, *** *p* < 0.001, and **** *p* < 0.0001.).

## 5. Conclusions

This study explored the use of liposomes as a delivery system for sea buckthorn extract to enhance intracellular uptake with potential therapeutic benefits. Two types of edge activators (sodium cholate—NaC—or sodium deoxycholate—NaDC) were used to generate nanocarriers. Both had an impact on their morphology and influenced cell behavior. NaC nanocarriers formed small and homogenous nanovesicles, with an average diameter of 25.8 nm in cryo-EM and with no cytotoxicity or impairment of fibroblast proliferation when studied in real time for up to 96 h. NaC- diluted hollow nanocarriers and phytocarriers (nanocarriers loaded with sea buckthorn extract) are uptaken by cells and induce an anti-inflammatory response in the presence of a low concentration of LPS (50 ng/mL). In conclusion, cell-compatible, therapeutically effective phytocarriers for sea buckthorn extract delivery were obtained and can be further tested in in vivo models.

## Figures and Tables

**Figure 1 pharmaceuticals-18-00212-f001:**
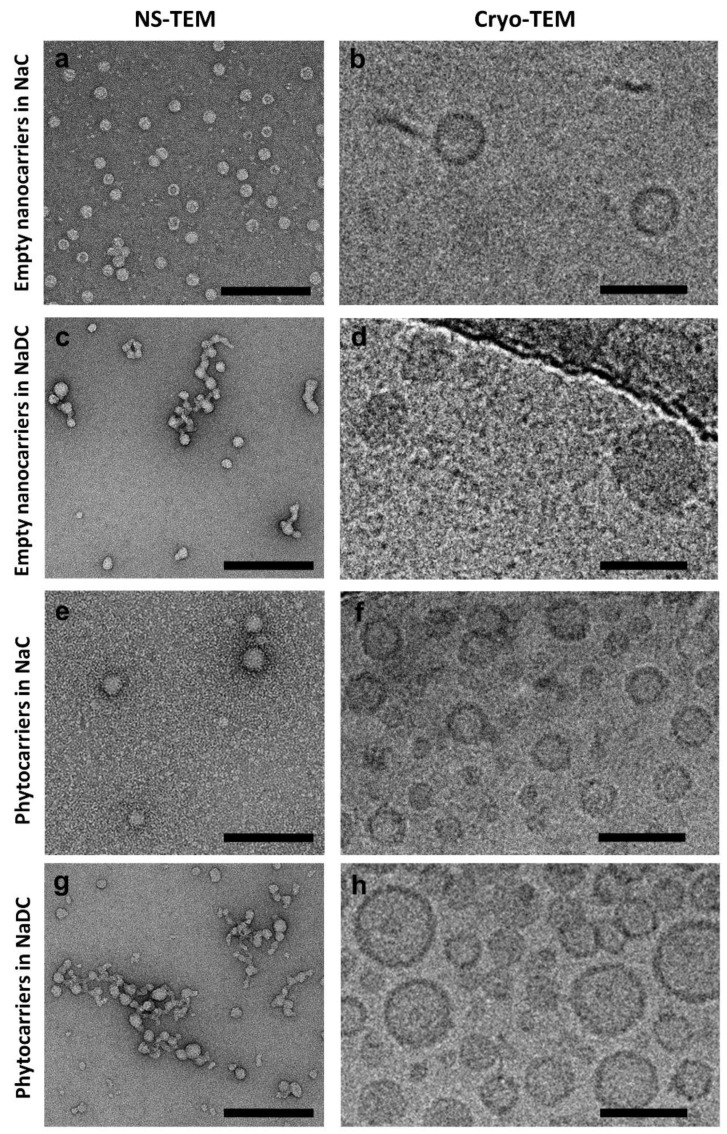
TEM analysis of phytocarriers. NaC hollow nanocarriers were relatively small and highly homogenous in size and shape in both NS-TEM (**a**) and cryo-TEM (**b**). By contrast, NaDC hollow nanocarriers were frequently aggregated and appeared polymorphic in NS-TEM (**c**), whereas individual particles were more electron-dense (darker) and with a comparatively broader size distribution in cryo-TEM (**d**). NaC phytocarriers were also smaller and more homogenous in both size and shape (**e**,**f**), compared to their NaDC counterparts (**g**,**h**). The scale bar is 200 nm for all NS-TEM images and 50 nm for all cryo-TEM images.

**Figure 2 pharmaceuticals-18-00212-f002:**
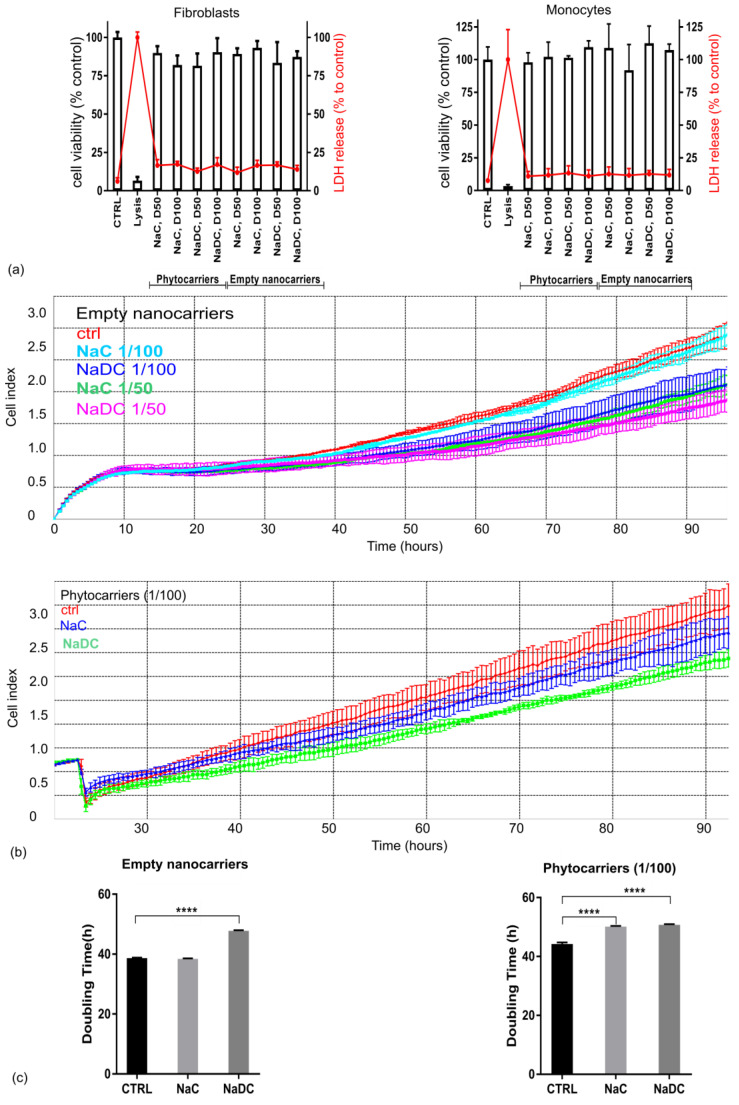
Viability and proliferation rates of cells treated with NaC and NaDC nanocarriers and phytosomes, assessed by endpoint assay (**upper panel**) and real-time assay (**lower panel**). (**a**) MTS bars and LDH points represent the average of triplicates ± S.D. (**b**) Real-time impedance graph points represent the average of triplicates ± S.D. Points were registered every 15 min. (**c**) Doubling times were calculated for 1/100 dilution, and the bars represent the average of triplicates ± S.D. One-way ANOVA, with Dunnet post hoc analysis, was used for the statistical analysis. **** *p* < 0.0001).

**Figure 3 pharmaceuticals-18-00212-f003:**
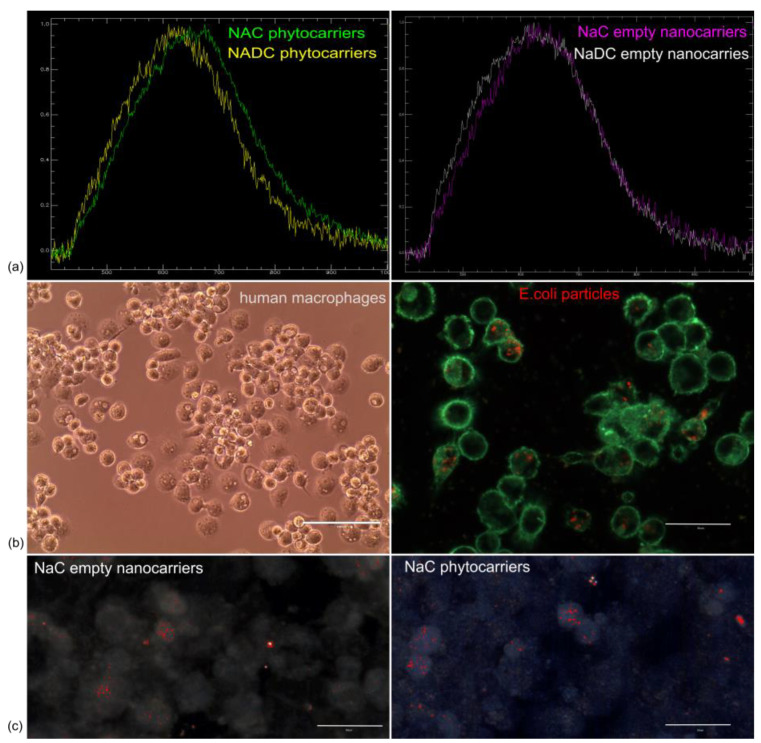
Phase-contrast and dark-field fluorescence hyperspectral imaging of phytocarriers. (**a**) Mean spectral analysis of phytocarriers obtained via hyperspectral analysis; (**b**) phagocytosis abilities of normal human macrophages investigated by phase-contrast microscopy (left, 40×) and fluorescence microscopy (60×), using fluorescently labeled E coli particles; (**c**) digitally highlighted overlayed HSI libraries of NaC phytocarriers and nanocarriers onto macrophage cell cultures (60×), which demonstrated their presence inside the cells. The focal plane was chosen based on the nucleus focal plane.

**Figure 4 pharmaceuticals-18-00212-f004:**
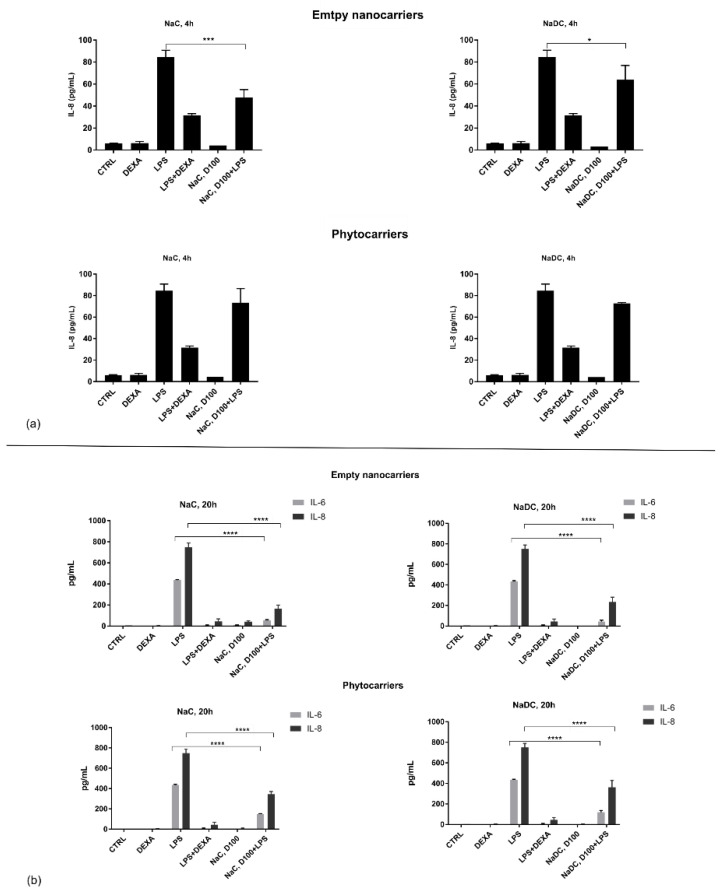
Dose-dependent pattern inhibition of IL-8 at 4 h (**a**) and of IL-6 and IL-8 at 20 h (**b**). One-way ANOVA, Dunnett’s post hoc analysis; * *p* < 0.05, *** *p* < 0.001, and **** *p* < 0.0001.

**Table 1 pharmaceuticals-18-00212-t001:** Zeta potential and polydispersity index of phytocarriers.

Nanocarriers	Zeta Potential (mV)	Polydispersity Index (PdI)
NaC hollow	−32 ± 6.7	0.283
NaC loaded	−23.9 ± 5.36	0.300
NaDC hollow	−38 ± 6.90	0.253
NaDC loaded	−36 ± 3.19	0.334

**Table 2 pharmaceuticals-18-00212-t002:** Composition profile of the sea buckthorn extract.

Composition	Concentration μg/mL
Gallic acid	17.93
Catechin	28.61
Epicatechin	45.77
Syringic acid	4.02
Coumaric acid	1.01
Rutin	46.25
Ellagic acid	7.65
Quercetin	6.38
Isorhamnetin	29.87

## Data Availability

The original contributions presented in this study are included in the article. Further inquiries can be directed to the corresponding author.

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
