# Peer review of "Potential of Newly Synthesized Sea Buckthorn Phytocarriers as Anti-Inflammatory Active Agents"

_pharmaceuticals, 2025, doi:10.3390/ph18020212_

Round 1

Reviewer 1 Report

Comments and Suggestions for Authors

The work is relevant and useful for the task. In its current form, it is devoid of obvious shortcomings and does not require significant changes. It was performed using modern analysis tools. The results are reliable and the main conclusions of the work are substantiated. An effective method of intracellular delivery was developed and the anti-inflammatory effect of phytocarriers based on sodium cholate and sodium deoxycholate loaded with sea buckthorn berry extract was studied. Cell-compatible, therapeutically effective phytocarriers for delivering sea buckthorn extract were obtained.

Apparently, the work has been significantly revised and the necessary changes have been made. Minor comments from the reviewer can be easily taken into account and the work can be recommended for publication.

1. The caption to Figure 1 states “… individual particles were more electron-dense…”. What does this mean? Explain.

2. The scale in Figures 2 a, b and c is too small to read. Increase the scale.

3. The terminology “… empty and loaded NaC nanocarriers and phytocarriers…” is used. Does the term “empty” really correspond to hollow “nanocarriers and phytocarriers”, the volume of which is then filled? If so, this should be stated more clearly.

However, Figure 1 does not state this clearly. Perhaps it would be better to use other terms (e.g. “… free and loaded NaC nanocarriers and phytocarriers…”) or add an explanation in the manuscript.

Author Response

Thank you for the time taken to assess our manuscript and your appreciative words.

Please find attached our reply, addressing point-by-point your suggestions.

Reviewer 2 Report

Comments and Suggestions for Authors

This study evaluates NaC and NaDC-based phytocarriers loaded with sea-buckthorn berry extract, which is rich in antioxidants and fatty acids. Using negative and electron cryo-microscopy, NaC phytocarriers were found to be spherical, while NaDC ones were polymorphic. Both types showed short-term biocompatibility, but NaDC delayed fibroblast proliferation in long-term assays. NaC phytocarriers were taken up by cells and demonstrated anti-inflammatory effects by reducing IL-8 production in monocytes after four hours. The study concludes that NaC-based phytocarriers are cell-compatible and have anti-inflammatory properties, making them promising for therapeutic use.

The manuscript is interesting because it looks at new ways to deliver drugs using natural materials. By creating phytocarriers from sea buckthorn, the research meets the growing need for safe and biodegradable options in medicine. The study's findings on the shape, safety, and anti-inflammatory effects of these carriers show they could help treat inflammatory conditions. The use of advanced microscopy techniques makes the results reliable and suggests these carriers could be useful not only in medicine but also in health and beauty products, expanding their potential uses.

A very well-planned study. The experimental part is described correctly and it is clear how the study was conducted. Well-designed and well-written. Typing errors are noticeable in places. Please revise your manuscript and correct them all.

Based on the above I would like to recommend the article be published after minor revision. 

Author Response

We thank the reviewer for the time spent on our manuscript and the positive evaluation of our work.

We have proofed the manuscript with TrackChanges, for ease of evaluation.

Reviewer 3 Report

Comments and Suggestions for Authors

The present study was well-designed and executed, yielding interesting, informative, and novel outcomes. However, there are some concerns that need to be addressed before proceeding further.

Comments to authors

1.      The detailed analysis of the morphological features of NaC or NaDC-loaded and empty nanocarriers was demonstrated using TEM imaging. However, it is recommended to study the size and uniformity of the nanocarriers using particle size analysis. This additional data would provide more meaningful insights into their size distribution.

2. Since sea-buckthorn extract was used as a phytocarrier, numerous studies have evidenced the presence of important phytochemicals (as mentioned by the authors). Therefore, it is important to provide the phytochemical profile of the extract used in this study. Various factors influence the composition of phytochemicals, and including this information in the revised version would add significant value.

3.      Additionally, data on the drug-loading capacity of the phytocarriers should be included to provide a comprehensive understanding of their functionality.

4.      There are some typographical errors that need to be corrected, such as the one in Line 221.

Author Response

We thank the reviewer for the time spent on our manuscript and the constructive suggestions.

We have added the suggested data and we addressed point-by-point all the concerns raised by the reviewer in the attached document.
